# An Abl-FBP17 mechanosensing system couples local plasma membrane curvature and stress fiber remodeling during mechanoadaptation

Asier Echarri[1]*, Dácil M. Pavón[1], Sara Sánchez[1], María García-García[1], Enrique Calvo[2], Carla Huerta-López[3], Diana Velázquez-Carreras [3], Christine Viaris de Lesegno[4], Nicholas Ariotti[5], Ana Lázaro-Carrillo[1,10], Raffaele Strippoli [6], David De Sancho [7,8], Jorge Alegre-Cebollada[3], Christophe Lamaze[4], Robert G. Parton [5,9] & Miguel A. Del Pozo [1]*

Cells remodel their structure in response to mechanical strain. However, how mechanical forces are translated into biochemical signals that coordinate the structural changes observed at the plasma membrane (PM) and the underlying cytoskeleton during mechanoadaptation is unclear. Here, we show that PM mechanoadaptation is controlled by a tension-sensing pathway composed of c-Abl tyrosine kinase and membrane curvature regulator FBP17. FBP17 is recruited to caveolae to induce the formation of caveolar rosettes. FBP17 deficient cells have reduced rosette density, lack PM tension buffering capacity under osmotic shock, and cannot adapt to mechanical strain. Mechanistically, tension is transduced to the FBP17 F-BAR domain by direct phosphorylation mediated by c-Abl, a mechanosensitive molecule. This modification inhibits FBP17 membrane bending activity and releases FBP17-controlled inhibition of mDia1-dependent stress fibers, favoring membrane adaptation to increased tension. This mechanoprotective mechanism adapts the cell to changes in mechanical tension by coupling PM and actin cytoskeleton remodeling.

[1] Mechanoadaptation and Caveolae Biology Laboratory, Cell and Developmental Biology Area, Centro Nacional de Investigaciones Cardiovasculares (CNIC), Melchor Fernández Almagro, 3, 28029 Madrid, Spain. [2] Proteomics Unit, Vascular Pathophysiology Area, Centro Nacional de Investigaciones Cardiovasculares (CNIC), Melchor Fernández Almagro, 3, 28029 Madrid, Spain. [3] Molecular Mechanics of the Cardiovascular System Laboratory, Cell and Developmental Biology Area, Centro Nacional de Investigaciones Cardiovasculares (CNIC), Melchor Fernández Almagro, 3, 28029 Madrid, Spain. [4] Membrane Mechanics and Dynamics of Intracellular Signaling Laboratory, Institut Curie – Centre de Recherche, PSL Research University, CNRS UMR3666, INSERM U1143, 75248 Paris, France. [5] The Institute for Molecular Bioscience, The University of Queensland, Brisbane, QLD 4072, Australia. [6] Department of Molecular Medicine, Sapienza University, Rome, Italy. [7] Departamento de Ciencia y Tecnología de Polímeros, Euskal Herriko Unibertsitatea, 20018 Donostia-San Sebastián, Spain. [8] Donostia International Physics Center, Manuel Lardizabal Ibilbidea, 4, 20018 Donostia-San Sebastián, Spain. [9] The Centre for Microscopy and Microanalysis, The University of Queensland, Brisbane, QLD 4072, Australia. [10]Present address: Departamento de Biología, Universidad Autónoma de Madrid, Cantoblanco, 28049 Madrid, Spain. *email: aecharri@cnic.es; madelpozo@cnic.es

Many physiological processes, including embryo development, wound healing, organ homeostasis, lipid storage and muscle activity, are exposed to various types of potentially damaging mechanical forces[1,2]. The plasma membrane (PM) acts as a protecting barrier that must accommodate the fluctuations in tension derived from mechanical action, which is essential to prevent its rupture upon excessive force. The proper adaptation of the cell to these tensional forces (i.e., cell mechanoadaptation) requires a tension-sensing mechanism ideally able to transduce this signal and initiate the changes needed to maintain cell and tissue integrity. PM remodeling is an essential step during this adaptation[3,4]. Concomitant to PM remodeling, stress fibers, which are tightly coupled to the PM[5,6], are highly sensitive to tension changes[7]. The tension-sensing pathways that remodel the PM and stress fibers during the adaptation to tension changes are still poorly understood.

The lipid bilayer is intrinsically fragile and evolution has developed complex mechanisms that contribute to prevent its rupture and to fix membrane wounds produced by excessive mechanical stress[8,9]. Local changes in the curvature of PM domains play an important role in preventing membrane rupture[4,10]. One of these membrane curvatures is exemplified by caveolae, which are small PM invaginations shaped by caveolins and cavins[11]. Caveolin1 (Cav1) and 3 are required to form caveolae in non-muscle and muscle cells, respectively, while cavin1 is needed in all tissues where caveolae are detected by electron microscopy (EM)[12,13]. In addition to these main caveolar components, other membrane curvature-generating molecules play a role in caveolae dynamics[14–18]. Reflecting a tight coupling with the cytoskeleton, caveolar dynamics are also highly dependent on stress fiber regulators, including formin mDia1 and Abl kinases[5,19]. Caveolae have the ability to flatten out in response to osmotic swelling and stretching, allowing the PM to reduce its tension, which prevents membrane rupture and protects cells exposed to mechanical insults, both in cultured cells and in vivo[4,20–23]. Mutations in *CAV1*, *CAV3*, and *CAVIN1* lead to muscular dystrophies, lipodystrophy, and other phenotypes, which may be explained at least in part by such mechanoprotective role of caveolae[9,24]. Interestingly, the signaling capacity of Cav3, in addition to its mechanoprotective role, is altered in myotubes expressing *CAV3* mutations found in muscular dystrophy patients[25]. Caveolae are frequently organized in clusters of different caveolar density that are connected with the PM through larger invaginations or shared necks; these structures are collectively named caveolar rosettes and are abundant in mechanically stressed tissues[19,26]. EHD proteins, recruited to the caveolar neck, have been recently shown to be involved in their formation[27].

Many PM remodeling activities, such as filopodia, lamellipodium extension, and endocytosis/exocytosis or membrane ruffles, are coupled to actin cytoskeleton reorganization[6]. In many of these processes, BAR proteins play an important role[28]. The BAR protein family is characterized by the presence of a BAR domain, which has an intrinsic curvature that forces the PM to bend[29–31]. Various proteins of this family regulate clathrin-dependent and -independent endocytosis[28,31–34]. The F-BAR subfamily member FBP17 (formin-binding protein 17) binds PIP2 and phosphatidylserine and oligomerizes through its N-terminal F-BAR domain, resulting in a strong membrane bending and tubulation activity[31,35,36]. Interestingly, FBP17 and its *Drosophila* homolog Cip4/Toca1 activate Arp2/3-dependent actin polymerization and inhibit the stress fiber regulator Diaphanous (mDia1–3 in mammals), respectively[35,37], highlighting the importance of these proteins in coordinating membrane remodeling and actin cytoskeleton dynamics. FBP17 directly binds mDia1[38], which is downstream of c-Abl in the pathway that links caveolae to stress fibers[5].

Here we identify FBP17 as a regulator of caveolar rosette assembly, PM tension adaptation, and stress fiber formation. In response to mechanical strain, FBP17-dependent membrane bending and stress fiber regulation are shut down by a direct inhibitory phosphorylation on its F-BAR domain by c-Abl kinase. C-Abl senses tension and possesses a mechanosensitive actin-binding domain that regulates its kinase activity needed to inhibit FBP17. Thus regulation of FBP17 by c-Abl allows a coordinated response of the PM and stress fibers to increased tension, which is important to mechanoprotect the cell.

## Results

**FBP17 favors the assembly of caveolar rosettes.** In order to identify proteins regulating caveolae biology, we screened a panel of candidates using a Cav1 inward trafficking assay. Upon loss of cell adhesion, a pool of PM-localized Cav1 moves from the PM to the endomembrane system in vitro and in vivo[39,40]. During this process, caveolar domains reorganize and clusters of caveolae are increased in the initial stages of the route[5]. During this reorganization of caveolar domains, membrane curvature is an obvious feature observed in EM images, not only in caveolae per se but also in the surrounding areas between caveolae of rosettes[11,40,41]. Although several caveolar components can induce local membrane curvature[17,42–44], we hypothesized that additional curvature regulators could be involved in regulating curvature locally in caveolar domains. The membrane curvature regulators of the BAR family[28,45] have already been linked, directly or indirectly, to caveolae[16,17,46,47]. Therefore, we screened various BAR proteins and used the Cav1 inward trafficking assay as a mean to test whether these proteins interfere with Cav1 and/or caveolae in any way.

We efficiently silenced pacsin2, SNX9, cip4, toca1, FBP17, and dynamin2 (positive control, Supplementary Fig. 1a). Pacsin2 inhibited the trafficking of Cav1 to the perinuclear area, in accordance with recently published results[48], validating our approach (Fig. 1a). SNX9, toca1, or cip4 silencing did not interfere with Cav1 trafficking. In contrast, FBP17 silencing blocked trafficking similar to dynamin2 and pacsin2 (Fig. 1a, Supplementary Fig. 1a). An additional small interfering RNA (siRNA) against FBP17 showed a similar effect (Fig. 1a). To confirm this result and to determine the stage in which FBP17 was acting, we stably silenced FBP17 using a different RNA interference target sequence in human fibroblasts (Fig. 1b). Quantification of the PM pool of endogenous Cav1 showed that in FBP17-silenced cells Cav1 moved away from the PM at a lower rate than control cells (Fig. 1c), suggesting a defect in the early stages of caveolae redistribution in detached cells.

Next, we studied the localization of endogenous FBP17. The antibody for FBP17 has been previously characterized, and we further assessed its specificity by absence of staining on FBP17 knockout (KO) cells (Supplementary Fig. 1b–e). In wild-type cells, a fraction of FBP17-positive spots either colocalized with or were adjacent to Cav1 spots (Fig. 1d). This fraction was reduced upon artificial lateral displacement of one of the channels, suggesting that the observed spatial correlation is specific (Fig. 1e). Cav1 and caveolae tend to drift and cluster upon disruption of the actin cytoskeleton[19], therefore if FBP17 was part of caveolar structures, disruption of the actin cytoskeleton should force FBP17 to co-cluster with Cav1 spots. Cytochalasin D (Cyt D) treatment induced co-clustering of Cav1 and FBP17 (Fig. 1d) as reflected by a significant increase in the proportion of FBP17 spots colocalizing with Cav1 spots (Fig. 1f).

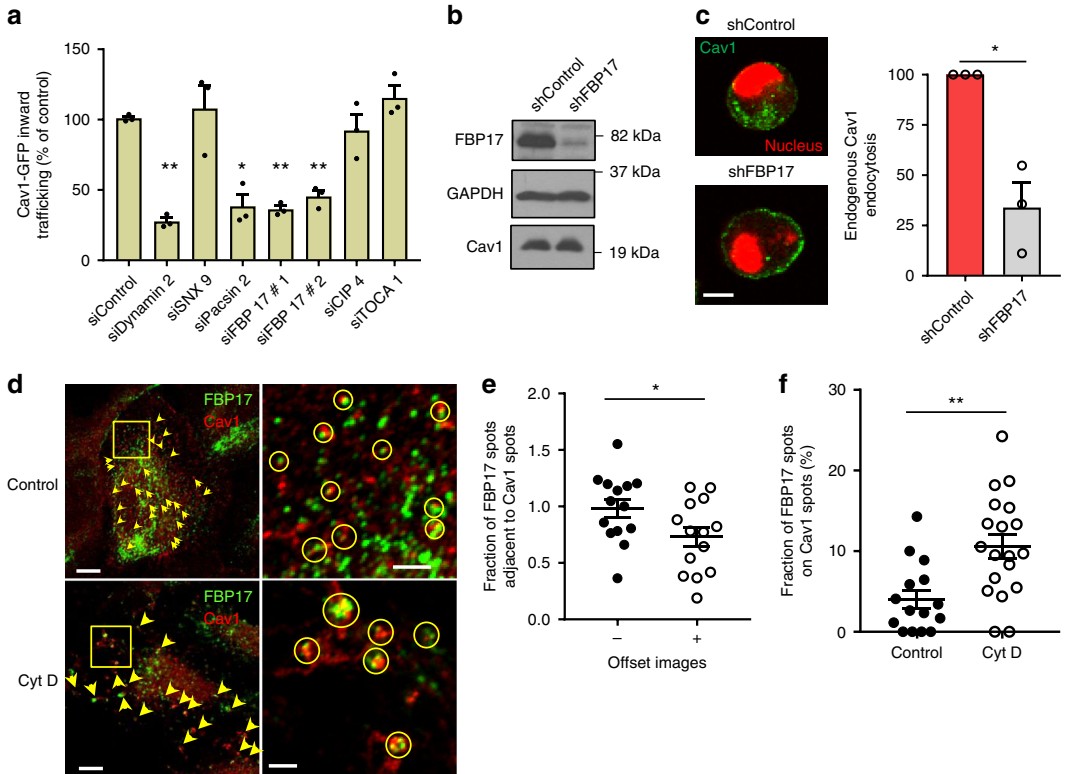

**Fig. 1 FBP17 regulates inward trafficking of Cav1 and localizes with Cav1. a** Depletion of FBP17, dynamin2 and pacsin2, but not other BAR proteins, blocks Cav1-GFP inward trafficking. HeLa cells expressing Cav1-GFP were transfected with the indicated siRNAs and were placed in suspension for 1 h, fixed, and perinuclear Cav1-GFP was scored. $N = 3$ biologically independent samples from 3 independent experiments. Statistical analysis with a two-tailed unpaired $t$ test. *$P < 0.05$; **$P < 0.01$. **b** Immunoblot showing specific lentivirus-mediated suppression of FBP17 in human fibroblasts. Equal levels of Cav1 and a loading control are shown. **c** Endogenous Cav1 is retained at the PM in FBP17 shRNA-silenced cells. The cell-edge endogenous Cav1 pool was normalized to the total amount of Cav1 in each cell. The reduction in cell-edge Cav1 from 5 to 20-min suspension time point was set as 100% disappearance from the PM in the control. Representative images of cells at the 20-min time point are shown. $N = 3$ biologically independent samples from 3 independent experiments. Statistical analysis with a two-tailed unpaired $t$ test. *$P < 0.05$. Scale bar 10 μm. **d** Immunofluorescence of endogenous Cav1 and FBP17. A fraction of both proteins exhibit an overlapping pattern or were adjacent to each other (marked in circles). Cyt D (1.25 μM for 60 min) and DMSO-treated cells were fixed and examined by confocal microscopy. Both molecules cluster together (marked in yellow circles and squares corresponding to the insets) in Cyt D-treated cells. Scale bar 10 and 2.5 μm (right panels). **e** Quantification of the overlapping/adjacent pattern between endogenous Cav1 and endogenous FBP17. The percentage of FBP17-positive spots overlapping with or adjacent to Cav1 spots was quantified in the original images (−offset image) and in images in which the image in one of the channels was offset (moved laterally) with respect to the other eight pixels (+offset image). The reduction in the overlapping pattern after lateral movement of one image indicates that the overlapping observed in the original image was not random. $N = 14$ biologically independent cells, representative of 3 independent experiments. 2850 (−) and 2861 (+) spots were scored. Statistical analysis with a two-tailed unpaired $t$ test. *$P < 0.05$. **f** Quantification of colocalization of endogenous Cav1 and FBP17 before and after Cyt D treatment (1.25 μM for 60 min). $N = 15$ (control) and $n = 19$ (Cyt D) biologically independent cells, representative of 3 independent experiments. Statistical analysis with a two-tailed unpaired $t$ test. **$P < 0.01$. Data represent mean ± S.E.M.

Similar results were obtained using full-length GFP-FBP17 and endogenous Cav1 (Supplementary Fig. 2a, b). Furthermore, live imaging of Cav1-mRFP+GFP-FBP17 or Cherry-FBP17+Cav1-GFP revealed that both molecules moved together, frequently in clusters (Supplementary Fig. 2c–f). Analysis of the localization of FBP17 by a recently developed EM-based high-resolution technique termed APEX-GBP[49] showed that some FBP17 localized around caveolar domains in single caveolae and in caveolar clusters containing tubular structures (Fig. 2a). Similarly, detailed analysis by immuno-gold EM (I-EM) showed that endogenous FBP17 was associated with caveolar domains but showed a clear preference for caveolar rosettes; while the majority of identified rosettes contained gold particles, only a small fraction of caveolae showed gold particles (Fig. 2b, c). FBP17 was preferably associated with the bulb of caveolae (Fig. 2d), either in the small fraction of labeled single caveolae or in caveolae within rosettes. Although some gold particles present in rosettes were far

from any visible caveolar bulb, the vast majority of gold was associated with the caveolar bulb (Fig. 2e).

We then analyzed whether FBP17 regulates the organization of Cav1. Endogenous Cav1 staining renders a mixed population of spots of varying intensity and size, likely reflecting different levels or caveolar organization[11,19]. Knockdown of FBP17 with two independent siRNAs (Supplementary Fig. 1a) significantly reduced the proportion of bright endogenous Cav1 spots, while that of dim spots was increased (Fig. 3a). The total level of Cav1 was not altered by FBP17 knockdown (Fig. 1b), suggesting that FBP17 specifically regulates the amount of caveolar clusters. Next, we quantified the number of caveolar structures by EM in control fibroblasts and fibroblasts silenced for FBP17. As shown in Fig. 3b, c, the number of rosettes, and caveolae associated with them, were significantly reduced in cells silenced for FBP17. Interestingly, the number of caveolae outside rosettes was unaffected upon FBP17 depletion (Fig. 3c), suggesting that

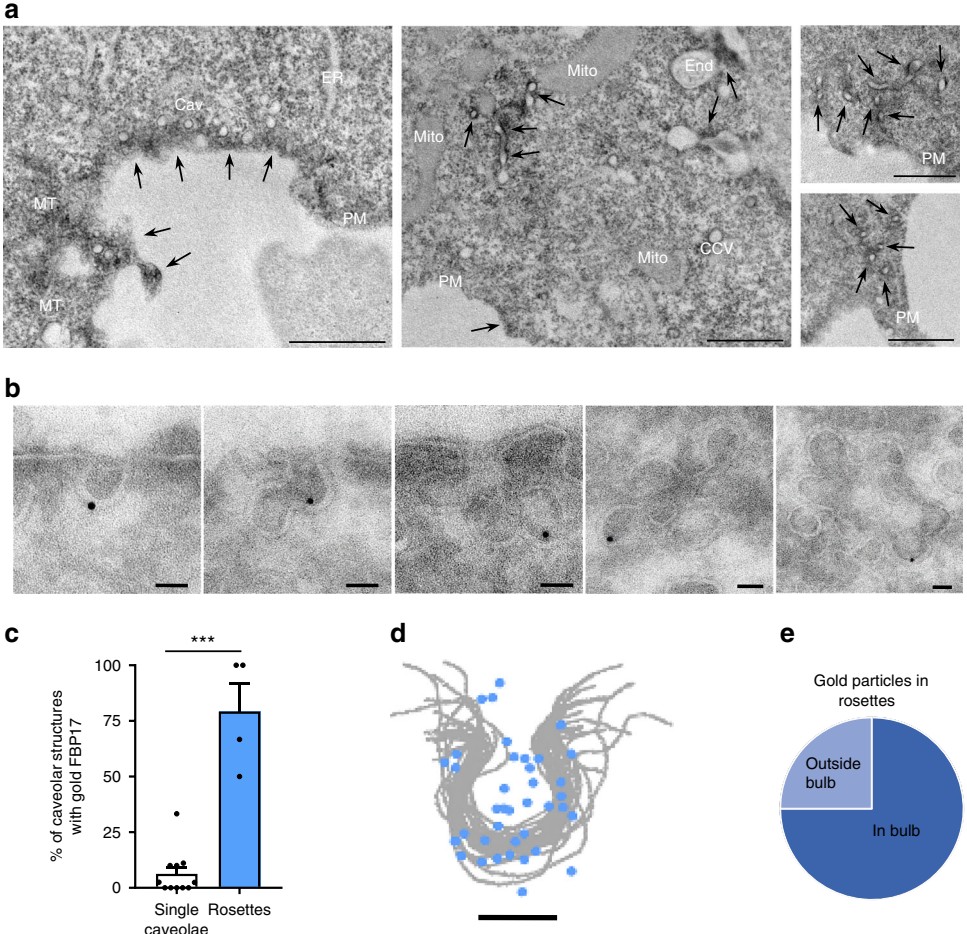

**Fig. 2 FBP17 is localized in caveolae and caveolar rosettes. a** EM of cells expressing GFP-FBP17. FBP17 signal (marked with arrows) is mostly located around caveolar clusters. Arrows indicate the presence of GFP-FBP17. Mito mitochondria, PM plasma membrane, Cav caveolae, End endosome, ER endoplasmic reticulum, MT microtubules, CCV clathrin-coated vesicle. Scale 1 μm. **b** Localization of endogenous FBP17 by I-EM. FBP17 (gold signal) in caveolae and in rosettes with different caveolar density are shown. Scale bar 50 nm. **c** Quantification of the fraction of caveolar structures with gold signal associated with FBP17 antibody. $N = 11$ (left) and 4 (right) biologically independent cells from 1 experiment. Data represent mean ± S.E.M. Statistical analysis with a two-tailed unpaired $t$ test. ***$P < 0.005$. **d** Localization of gold particles associated with endogenous FBP17 in caveolae. Scale bar 50 nm. **e** Fraction of gold particles associated with endogenous FBP17 in the indicated regions of rosettes.

FBP17 is specifically involved in the formation of rosettes. Rosettes were surface-connected as they were positive for ruthenium red (Supplementary Fig. 3a), which labels surface-connectivity. Depending on the section of the cut, rosettes may appear in the cell interior, with or without a visible neck (Supplementary Fig. 3a). Rosettes contain at least two differentiated structures, the large invagination and the caveolae that are formed around this invagination (Supplementary Fig. 3a). The large invaginations may (in which case we term them "rosettes") or may not contain caveolae ("large invagination," Supplementary Fig. 3a). Since the number of rosettes decreases upon FBP17 silencing, it could be that either FBP17 is required for caveolae formation exclusively in rosettes (in which case we should observe an increase in large invaginations in cells silenced for FBP17, proportional to the decrease in rosettes) or FBP17 is involved in the formation of the large invagination that holds caveolae, and therefore indirectly caveolae within, in which case we should observe either a similar or lower number of large invaginations in cells silenced for FBP17. Quantification of surface-connected large invaginations showed that these were similar in the absence of FBP17 (shControl 1.82 ± 0.4 and shFBP17 1.75 ± 0.3/100 μm PM, $t$ Student $p$ value 0.89), suggesting that FBP17 is specifically required for the formation

of rosettes but is not required for the large invaginations without caveolae. To further confirm these results, we knocked out the FBP17 in RPE-1 cells (Fig. 3d). Analogous results were observed in this system: rosettes were significantly reduced, while caveolae outside rosettes were unaffected, and total caveolae density was slightly reduced (Fig. 3e, f). Next, we asked what happens to Cav1 pools localized to caveolar rosettes when these are impaired upon FBP17 depletion. I-EM showed that Cav1 signal was (i) increased in caveolae outside rosettes and PM, (ii) reduced in rosettes and surface-connected vesicles, and (iii) unaffected at other cellular regions (Supplementary Fig. 3b–d), suggesting that, upon rosette disappearance due to reduced FBP17 levels, Cav1 redistributes to the PM and caveolae outside rosettes. Taken together, these observations support the hypothesis that FBP17 is specifically required for the formation of caveolar rosettes.

**FBP17 is mechanoprotective and regulates PM tension.** Caveolae protect cells from osmotic swelling[4]. Thus we investigated whether FBP17 contributes to cell mechanoprotection given its requirement for rosette formation. Cells knocked down for FBP17 were more sensitive to tension increase triggered by cell swelling and were damaged more easily than control cells,

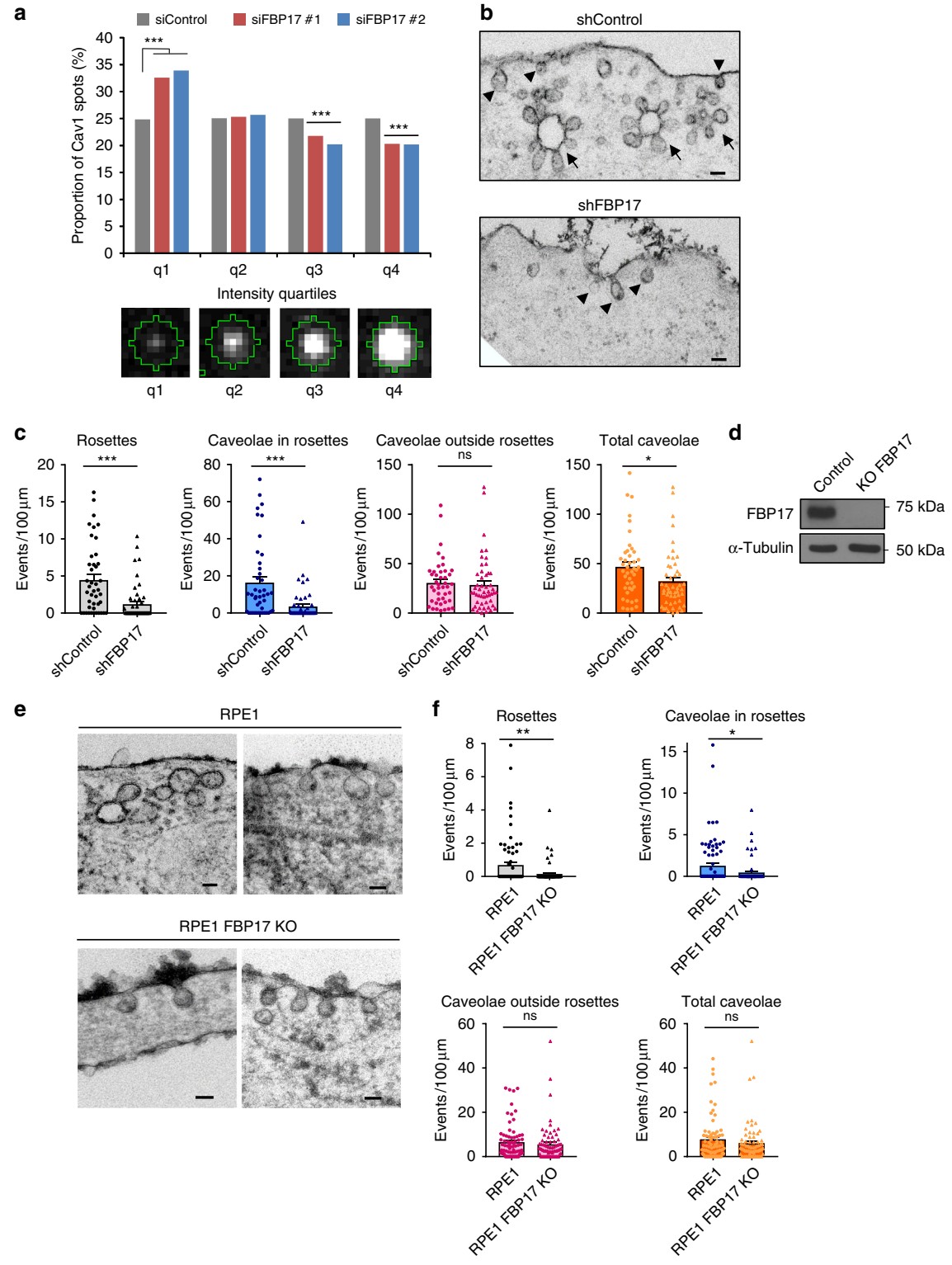

independently of the length of the treatment (10 or 2 min) (Fig. 4a, Supplementary Fig. 3e). Similarly, fibroblasts knocked down for FBP17 were more sensitive to prolonged mechanical stretching treatments (Fig. 4b), albeit effects induced by short treatments (30 min) were not significant (Supplementary Fig. 3f). Of note, neither FBP17 transcript nor protein levels were affected by prolonged mechanical stretching (Supplementary Fig. 3g, h). These results suggest that FBP17 is mechanoprotective.

Although rosettes are known to disassemble upon osmotic swelling[20], whether the large invagination of the rosette is also

mechanosensitive is not known. We analyzed the kinetics of PM remodeling upon osmotic swelling. The disassembly of caveolae was fast and reversible, as published[4] (Fig. 4c–g); on the contrary, the disassembly of rosettes, was biphasic, i.e., first caveolae within rosettes were disassembled (by 2 min) but the large invagination disassembled in a second phase (10 min) (Fig. 4c–g). Caveolae within rosettes were more sensitive to osmotic shock as compared to caveolae outside of rosettes, in accordance with previous observations[4,5,20]. Importantly, these rearrangements are reversible because rosettes, caveolae, and large invaginations were

**Fig. 3 FBP17 is needed for caveolar rosette formation. a** FBP17 induces the formation of large caveolar clusters. Endogenous Cav1 clustered spots are decreased in FBP17-silenced cells with two independent siRNAs. The Cav1 spot intensity values separated into four populations ($q_n$, $q_1$ being the dimmest spots and $q_4$ the brightest spots) are shown. A diagram of representative segmented Cav1 spots with different intensities is shown for clarity. $N = 12,948$ (shControl), $n = 14,144$ (shFBP17 #1), and $n = 11,570$ (shFBP17 #2) biologically independent spots, representative of three independent experiments. Chi$^2$ test ***$P < 10^{-5}$. **b** FBP17 is required for the formation of rosettes. Human fibroblasts silenced for FBP17 and control cells were processed for EM in the presence of ruthenium red to mark surface-connected structures. Caveolar rosettes (arrows) and caveolae outside rosettes (arrow head) are marked. Scale bar 100 nm. **c** The amount of rosettes and caveolae outside rosettes were counted in each condition in cells treated as in **b**. $N = 43$ (shControl) and 50 (shFBP17) biologically independent cells from 3 independent experiments. Statistical analysis with a two-tailed unpaired $t$ test. *$P < 0.05$; ***$P < 0.005$; ns non-significant. **d** Immunoblot of RPE-1 cells knocked out for FBP17. **e** Human RPE-1 KO for FBP17 and control cells were processed for electron microscopy in the presence of ruthenium red to mark surface-connected structures. Scale bar 100 nm. **f** The density of caveolar domains was counted in each condition in cells treated as in **e**. $N = 74$ (RPE1) and 72 (RPE1 FBP17 KO) biologically independent cells from 3 independent experiments. Statistical analysis with a two-tailed unpaired $t$ test. *$P < 0.05$; **$P < 0.01$. Data represent mean ± S.E.M.

reassembled after removal of the hypo-osmotic challenge (Fig. 4c–g). Disappearance of rosettes was not due to endocytosis as determined by I-EM, because the signal associated with caveolar domains upon hypo-osmotic shock decreased for Cav1 and remained constant in the case of FBP17 but increased for both proteins in PM regions without curvature (Supplementary Fig. 3i, j). Furthermore, the signal outside caveolar domains and PM did not increase (Supplementary Fig. 3i, j). Such enrichment specifically at non-caveolar regions of the PM, while excluding intracellular compartments, suggests that any redistribution of Cav1 or FBP17 induced by hypo-osmotic shock is not likely derived from endocytic mechanisms.

Based on these results, we hypothesized that FBP17 was important to control PM tension variations. As shown in Fig. 4h, cells knocked down for FBP17 had increased tension at the PM after osmotic shock, consistent with their higher sensitivity to hypo-osmotic shock and cell stretching and their significant reduction in rosettes (Fig. 3). In order to determine whether FBP17 regulates membrane tension variations and sensitivity to mechanical stress exclusively through caveolae, we deleted FBP17 in Cav1 KO cells (double KO cells; Supplementary Fig. 4a). In double KO cells, membrane tension buffering under osmotic shock was similar to Cav1 KO cells and the sensitivity to mild hypo-osmotic shock (60 mOsm) was also similar in both cell lines (Fig. 4i, j). However, when osmolarity was further reduced (30 mOsm), double KO cells were more sensitive to osmotic swelling than parental Cav1 KO cells (Supplementary Fig. 4b). Although PM tension changes could not be measured at this osmolarity due to technical limitations, these results suggest that FBP17 regulates additional pathways independent from caveolae, at least under low osmolarity conditions. EM analysis of double KO cells did not show any differences in large invaginations (Cav1 KO MEFs $0.3 \pm 0.12/100\,\mu\text{m}$ PM and double KO $0.34 \pm 0.14/100\,\mu\text{m}$ PM, $t$ Student $p$ value 0.83), suggesting that other activities of FBP17 were contributing to the mechanoprotection of the cell.

FBP17 has been shown to be sensitive to osmotic swelling[50]; we confirmed this sensitivity by osmotic swelling and determined that this effect was reversible after shock release (Supplementary Fig. 4c). Osmotic swelling may have additional effects such as dilution of the proteins; however, increased tension exerted by mechanical stretching of cells also reduced the number of FBP17 tubules (Fig. 4k), indicating that FBP17-generated membrane tubules are sensitive to mechanical stress.

**A kinase-dependent mechanism translates tension into FBP17.** To gain further mechanistic insight into how FBP17 senses tension and contributes to the membrane remodeling observed during the adaptive response to mechanical strain, we explored potential underlying sensing molecular mechanisms. Although it has been shown that FBP17 per se can sense PM tension

in vitro[50], we observed that endogenous FBP17 was phosphorylated on tyrosine residues after osmotic swelling in cells (Fig. 5a), which could be related to the change in its tubulation activity observed using the same conditions. This phosphorylation occurred at 2 min of treatment and remained constant even after shock release (Supplementary Fig. 5a, b). c-Abl tyrosine kinase regulates caveolae organization and trafficking and other BAR proteins are phosphorylated by c-Abl[51,52]. We tested whether c-Abl phosphorylates FBP17. In cells, exogenously expressed c-Abl strongly phosphorylated FBP17 but not an unrelated control protein (Fig. 5b). Most importantly, endogenous FBP17 phosphorylation was completely inhibited by c-Abl inhibition (Fig. 5c, d). Purified c-Abl was also capable of phosphorylating efficiently a GST-FBP17 fusion protein in vitro but not glutathione S-transferase (GST; Fig. 5e). Moreover, immunoprecipitation of GFP-c-Abl from cell lysates, but not green fluorescent protein (GFP) control, co-immunoprecipitated myc-FBP17 (Fig. 5f) and endogenous FBP17 co-immunoprecipitated with endogenous c-Abl (Fig. 5g). Moreover, both proteins significantly colocalized in cells (Fig. 5h, i).

If c-Abl-induced phosphorylation is involved in the transduction of increased tension and the regulation of PM remodeling, then c-Abl expression might inhibit FBP17-dependent membrane-bending activity. As shown in Fig. 6a–b, c-Abl overexpression strongly reduced the amount of GFP-FBP17 tubules, without affecting FBP17 protein levels (Fig. 6c). In order to determine whether c-Abl-induced phosphorylation was directly responsible for the reduced tubulation activity of FBP17, we identified by mass spectrometry the specific FBP17 tyrosine residues phosphorylated by c-Abl in vitro. This analysis identified residues Y190, Y205, and Y287 as major phosphorylation sites. Mutation of these residues to phenylalanine (FBP17 3YF) significantly reduced osmotic swelling-induced FBP17 phosphorylation in vivo (Fig. 6d, e) and phosphorylation by c-Abl in vitro (Supplementary Fig. 5c, d). Since phosphorylation of endogenous FBP17 was fully dependent on endogenous c-Abl (Fig. 5c, d), taken together, these results suggest that FBP17 is phosphorylated by c-Abl in cells on those three residues in response to osmotic swelling. Interestingly, these three residues are located within the F-BAR domain of FBP17, which could explain the inhibitory effect of c-Abl expression over FBP17 tubulation activity. To test this possibility, we mutated the three tyrosine residues to glutamic acid, to create a phosphomimetic mutant. FBP17 3YE distributed diffusely throughout the cytosol and was virtually unable to form any tubule or puncta, contrary to the wild-type form, either in wild-type cells (Fig. 6f, g) or in FBP17 KO reconstituted cells (Supplementary Fig. 5e, f). As expected, the non-phosphorylatable FBP17 mutant 3YF produced more tubules than the wild-type form (Supplementary Fig. 5g) and was insensitive to the inhibitory effect of co-expressed c-Abl (Supplementary Fig. 5h). Because the expression levels of

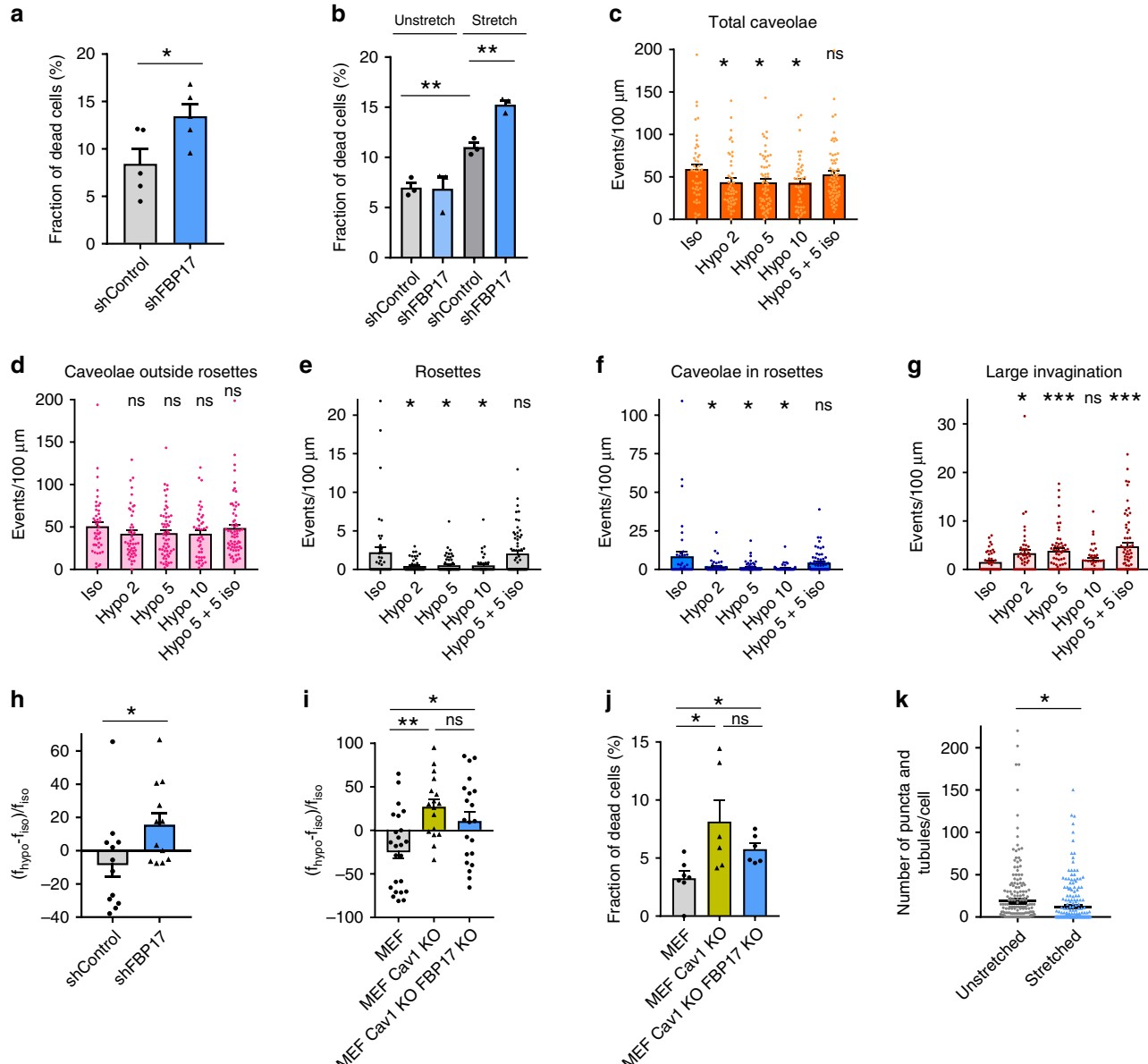

**Fig. 4 FBP17 is mechanoprotective and buffers PM tension under osmotic swelling. a** FBP17 expression was silenced in human fibroblast and cells were treated with hypo-osmotic medium (10 min). Trypan blue-labeled cells were scored and the fraction of dead cells was calculated. $N = 5$ biologically independent samples from 5 independent experiments. Statistical analysis with a two-tailed unpaired $t$ test. *$P < 0.05$. **b** FBP17-depleted human fibroblasts were subjected to stretching. Scored as in **a**. $N = 3$ biologically independent samples from 3 independent experiments. Statistical analysis with a two-tailed unpaired $t$ test. **$P < 0.01$. **c–g** Effect of the indicated treatments in caveolar structures. Quantification of the indicated structures observed by EM of human fibroblasts under hypo-osmotic shock for different time points (in minutes) and after shock release (back to iso-osmotic conditions, 5+5 iso) is shown. $N = 45$ (iso), 47 (2 hypo), 58 (5 hypo), 43 (10 hypo), and 65 (5+5 iso) biologically independent cells from 3 independent experiments. Statistical analysis with a two-tailed unpaired $t$ test. *$P < 0.05$; ***$P < 0.005$. **h, i** PM tension was measured using magnetic tweezers in different cell lines. Change of the mean tether force after hypo-osmotic shock is shown. $N = 12$ biologically independent cells from 12 independent experiments (**h**) and $n = 24$ (MEF), $n = 17$ (MEF Cav1 KO), and $n = 21$ (MEF Cav1 KO FBP17 KO) biologically independent cells from the corresponding independent experiments (**i**). Statistical analysis with a two-tailed unpaired $t$ test. *$P < 0.05$; **$P < 0.01$. **j** Cells were treated with hypo-osmotic medium and the fraction of trypan blue-labeled cells were scored. $N = 7$ (MEF), $n = 6$ (MEF Cav1 KO), and $n = 6$ (MEF Cav1 KO FBP17 KO) biologically independent experiments. Statistical analysis with a two-tailed unpaired $t$ test. *$P < 0.05$. **k** FBP17 tubulation activity is sensitive to stretching. HeLa cells were transfected with GFP-FBP17 and stretched or kept without stretching. The cells were fixed and the amount of FBP17-labeled tubules was scored. $N = 224$ (unstretched) and $n = 226$ (stretched) biologically independent cells from 3 independent experiments. Statistical analysis with a two-tailed unpaired $t$ test. *$P < 0.05$. Data represent mean ± S.E.M. ns non-significant.

FBP17 affect its associated tubulation phenotype, protein levels of the different FBP17 constructs were carefully controlled to ensure reliable comparison (Supplementary Fig. 5i–k). In addition, EM analysis confirmed that the 3YE mutant, as opposed to the wild-type variant, was not associated with caveolar domains or any other identifiable structure and distributed diffusely throughout the cytosol (Supplementary Fig. 5l). These observations suggested that phosphorylation of these tyrosine residues, localized in the F-BAR domain, directly impacts on the membrane-bending activity of FBP17. Further supporting this notion, while purified FBP17

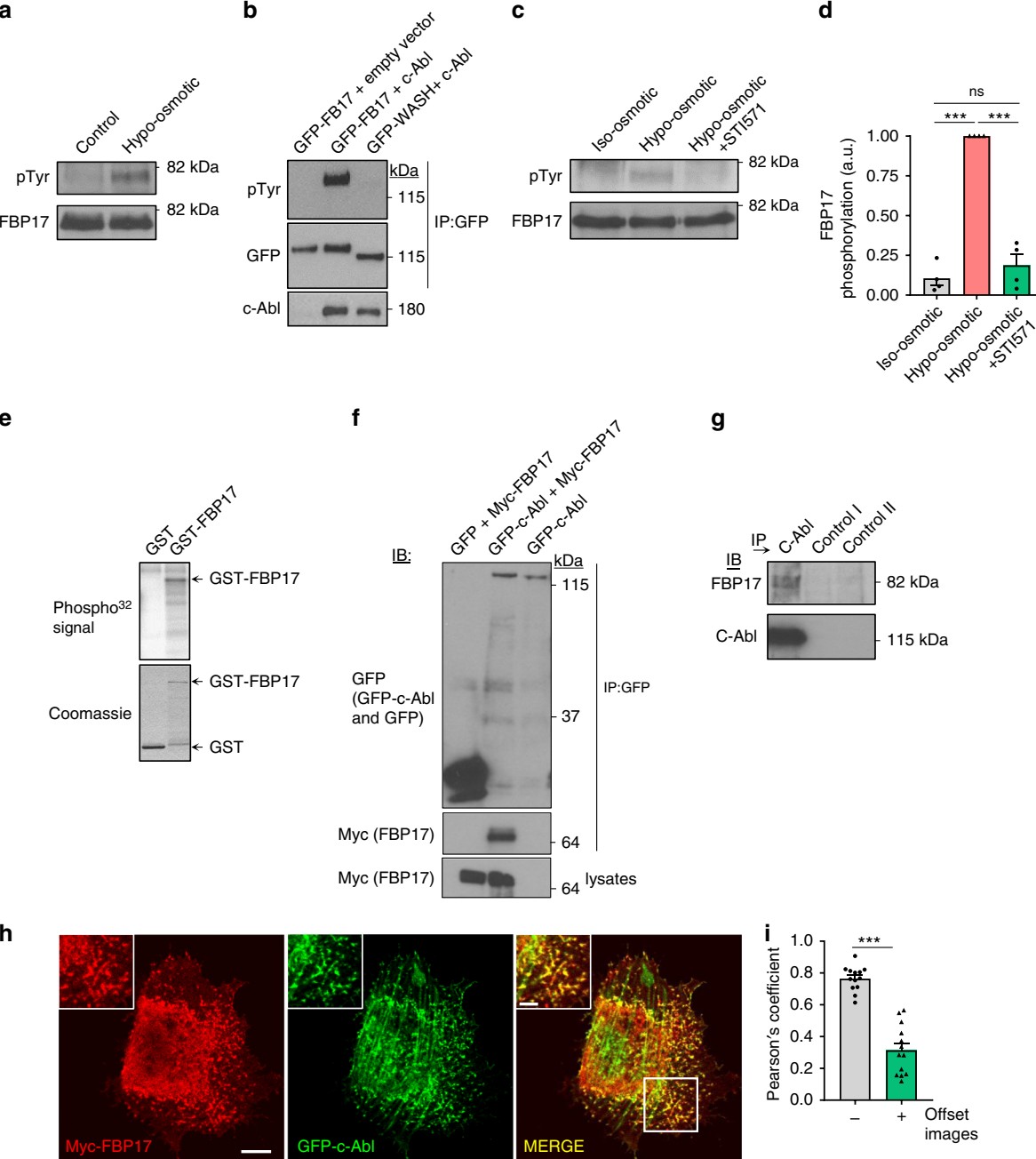

**Fig. 5 FBP17 is phosphorylated by osmotic swelling and c-Abl tyrosine kinase. a** Human fibroblast were treated with hypotonic or isotonic medium for 10 min and endogenous FBP17 was immunoprecipitated. Phosphotyrosine (pTyr) and total FBP17 were identified by western blot. **b** The indicated proteins were expressed in cells and GFP fusion proteins were immunoprecipitated and the phospho-tyrosine signal detected with antibodies against phospho-tyrosine (Ptyr, with 4G10 antibody). **c** Human fibroblast were treated with hypotonic or isotonic medium for 10 min and endogenous FBP17 was immunoprecipitated. Abl inhibitor (STI571, 10 μm) was preincubated 30 min before the treatment and added during the treatment. Phosphotyrosine (pTyr) and total FBP17 were identified by western blot. **d** Quantification of **c**. $N = 4$ biologically independent samples from 4 independent experiments. Statistical analysis with a two-tailed unpaired $t$ test. ***$P < 0.005$; ns non-significant. **e** Pure c-Abl kinase phosphorylates pure FBP17 in vitro. GST and GST-FBP17 were purified and 0.5 μg was incubated with c-Abl. The total amount of protein and phosphorylated radioactive signals are shown. **f** c-Abl interacts with FBP17. GFP-c-Abl and Myc-FBP17 were transfected into 293T cells and c-Abl was immunoprecipitated using anti-GFP antibody. Immunoblots (IB) of the immunoprecipitates (IP) and whole-cell lysates are shown. **g** Endogenous c-Abl interacts with endogenous FBP17. c-Abl was immunoprecipitated and FBP17 was identified in the IP but not in the IP of two control antibodies. Immunoblots of the immunoprecipitated proteins are shown. **h** HeLa cells grown on coverslips were transfected with GFP-c-Abl and myc-FBP17, stained, and examined by confocal microscopy. Scale bar 10 μm, inset 4 μm. **i** Quantification of **h**. Quantification of the colocalization between expressed c-Abl and FBP17. The images were artificially shifted (offset+) 8 pixels from each other and Pearson's coefficient was calculated again, to rule out random signal overlap. $N = 13$ biologically independent cells, representative of 3 independent experiments. Statistical analysis with a two-tailed unpaired $t$ test. ***$P < 0.005$. Data represent mean ± S.E.M.

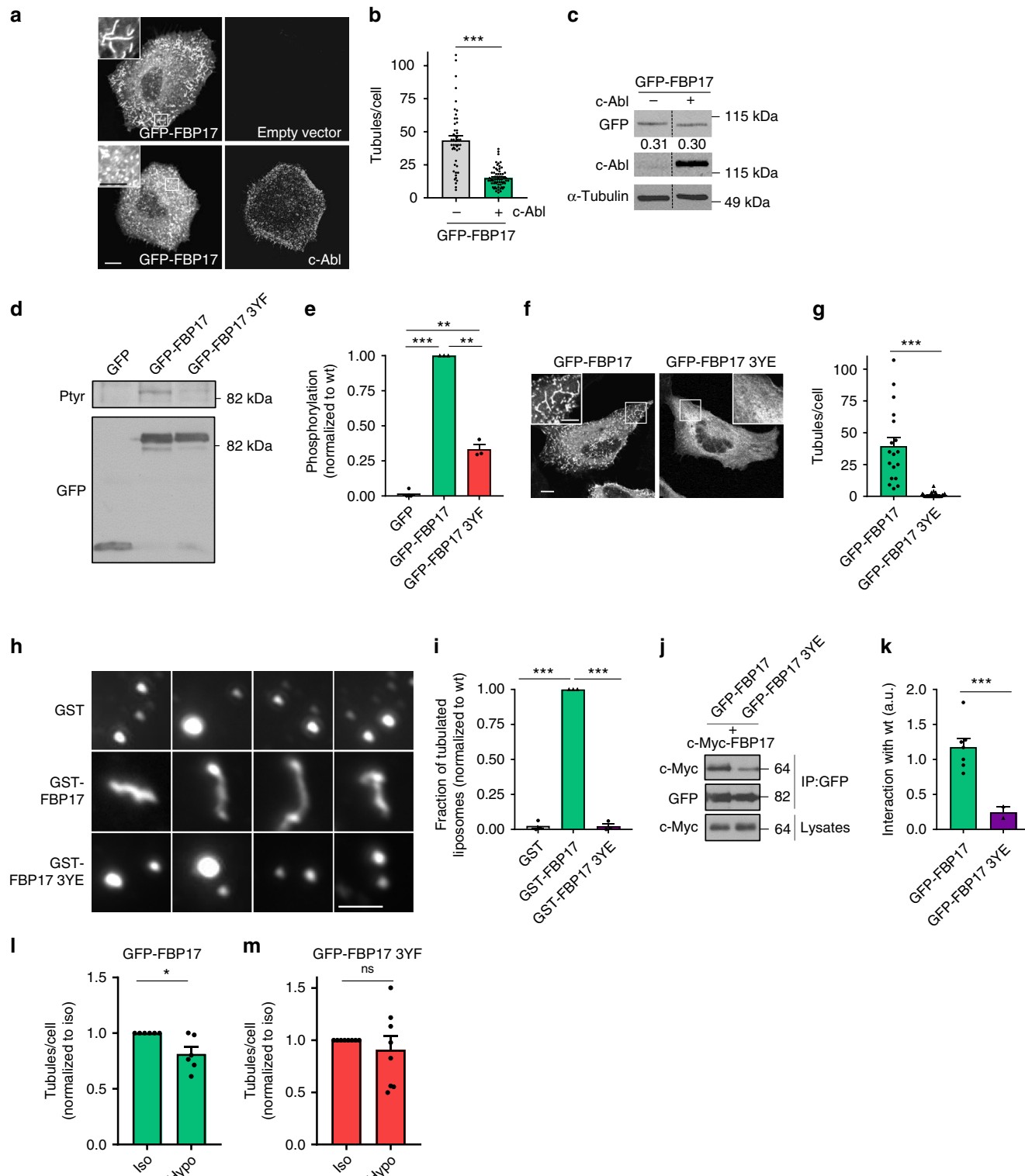

efficiently induced tubulation of liposomes in vitro, the 3YE mutant did not (Fig. 6h, i). Because dimerization of many BAR proteins, including FBP17, is a required property for their membrane tubulation activity[36,53], we assessed the dimerization capacity of FBP17 3YE through co-immunoprecipitation assays. The 3YE mutant exhibited a weak interaction with the wild-type protein, as compared to the robust co-immunoprecipitation of the wild-type/wild-type protein pair (Fig. 6j, k). Thus a direct consequence of c-Abl-mediated phosphorylation in the FBP17 F-BAR domain is the inhibition of its membrane-bending activity.

To determine whether phosphorylation is important to inhibit FBP17 membrane-bending activity in vivo in response to tension increase, we compared the membrane tubulation activity of wild-type FBP17 and 3YF, which is not phosphorylatable, under different hypo-osmotic conditions. At low osmolarity conditions (30 mOsm), no significant differences were observed between both proteins (Supplementary Fig. 5m, n). However, the 3YF mutant was insensitive to milder osmotic shock (60 mOsm), as compared to cells expressing the wild-type FBP17 (Fig. 6l, m). Thus phosphorylation is important to control FBP17 activity in cells in response to tension increase.

**Fig. 6 Tension increase is translated into an inhibitory phosphorylation in the FBP17 F-BAR domain. a, b** c-Abl inhibits FBP17 tubulation activity. c-Abl or empty vector were expressed with GFP-FBP17 in HeLa cells and c-Abl was stained and examined with confocal microscopy. $N = 64$ (−) and $n = 47$ (+) biologically independent cells, representative of 3 independent experiments. Statistical analysis with a two-tailed unpaired $t$ test. \*\*\*$P < 0.005$. Scale bar 10 μm, inset 5 μm. **c** c-Abl expression does not affect the expression level of GFP-FBP17. Immunoblot showing the levels of GFP-FBP17 and c-Abl and loading control. The relative amount of GFP-FBP17 in each condition is indicated in arbitrary units. **d, e** GFP-FBP17 and GFP-FBP17 3YF were expressed in HeLa cells and the amount of phosphorylated FBP17 in response to osmotic swelling was determined. $N = 3$ biologically independent samples from 3 independent experiments. Statistical analysis with a two-tailed unpaired $t$ test. \*\*$P < 0.01$; \*\*\*$P < 0.005$. **f, g** GFP-FBP17 and GFP-FBP17 3YE were expressed in HeLa cells and the amount of tubules per cell was calculated. $N = 18$ and $n = 24$ biologically independent cells, representative of 3 independent experiments. Statistical analysis with a two-tailed unpaired $t$ test. \*\*\*$P < 0.005$. Scale bar 10 μm, inset 5 μm. **h, i** Pure GST-FBP17, GST-FBP17 3YE, and control GST were incubated with liposomes and the amount of tubulated liposomes were scored. $N = 3$ biologically independent samples, from 3 independent experiments. Statistical analysis with a two-tailed unpaired $t$ test. \*\*\*$P < 0.005$. Scale bar 2.5 μm. **j, k** FBP17 3YE is unable to interact with FBP17 wild type. GFP-tagged FBP17 wild type of 3YE mutant were expressed in cells together with c-Myc-FBP17. GFP versions were immunoprecipitated and their binding to wild-type c-Myc-FBP17 was quantified. Numbers in the right margin indicate kDa. $N = 7$ and $n = 2$ biologically independent samples from the corresponding independent experiments. Statistical analysis with a two-tailed unpaired $t$ test. \*\*\*$P < 0.005$. **l, m** HeLa cells were transfected with GFP-FBP17 or GFP-FBP17 3YF and exposed to osmotic swelling (60 mOsm) for 10 min. The amount of FBP17-labeled tubules was scored. $N = 6$ (**l**) and $n = 8$ (**m**) biologically independent cells from the corresponding independent experiments. Statistical analysis with a two-tailed unpaired $t$ test. \*$P < 0.05$; ns non-significant. Data represent mean ± S.E.M.

**C-Abl kinase is mechanosensitive.** Because c-Abl-dependent phosphorylation of FBP17 is an event triggered by PM tension increases and c-Abl is associated with the PM and actin cytoskeleton[54], we wondered whether c-Abl possesses intrinsic mechanosensing activity. In order to test this hypothesis, we first assessed whether c-Abl activity is regulated by mechanical stimuli. c-Abl kinase activity was significantly increased in cells exposed to hypo-osmotic swelling, either at 60 or 30 mOsm (Fig. 7a, b). Activation occurred relatively fast as it was already detectable 2 min after the shock and remained constant after shock release, consistent with FBP17 phosphorylation kinetics (Fig. 7b). Furthermore, endogenous c-Abl activity was elevated by a stiff substrate and mechanical stretching (Fig. 7c, d), suggesting that c-Abl is regulated by tension changes from different forms of mechanical stimuli (Fig. 7e). This sensitivity to mechanical stimuli could be due to the intrinsic mechanosensitive properties of c-Abl or to its activation by a mechanosensitive upstream molecule. A clear candidate to transduce mechanical stimuli to c-Abl was Src, but c-Abl was equally activated by osmotic shock in Src/Fyn/Yes KO cells (Supplementary Fig. 6a). Mechanosensitive proteins frequently bind F-actin, such as talin, myosin, or filamin A[55]. c-Abl binds F-actin, and binding to F-actin is important to regulate its kinase activity in vitro and in vivo[56]. Interestingly, the crystal structures of the actin-binding domains (ABDs) of c-Abl and Abl2 show strong structural homology to structural domains of mechanosensitive molecules, such as Talin, vinculin, FAK, and α-catenin[57,58]. Specifically, the ABD domain of c-Abl and Abl2 show the strongest structural homology to mechanosensitive domains of talin, which have been shown to unfold at low forces leading to exposure of cryptic-binding sites[57–59]. Thus we hypothesized that the ABD domain of c-Abl could be sensitive to mechanical forces, allowing c-Abl to sense mechanical stimuli and regulate its activity. In order to test this hypothesis, we stretched the ABD domain of c-Abl using single-molecule force spectroscopy by atomic force microscopy (AFM)[60] (Fig. 7f, g). To be able to fingerprint single-molecule events, the ABD domain was fused to four titin I91 domains, whose mechanical properties are well characterized[61]. The resulting heteropolyprotein, which retains the F-actin-binding function of ABD (Supplementary Fig. 6b), was pulled at constant velocity (force-extension, Fig. 7g) or at monotonically increasing forces (force-ramp, Supplementary Fig. 6c). We analyzed full-length single-molecule recordings in which the four I91 domains unfold, which ensures that the full-length heteropolyprotein, including the ABD domain, was under mechanical force. In force-extension traces, I91 unfolding events result in force peaks that correspond to changes in contour length

of 28–29 nm, as estimated using the worm-like chain of polymer elasticity (Fig. 7g)[62]. In force-ramp recordings, I91 unfolding events are detected as stepwise changes in length of 24–25 nm (Supplementary Fig. 6c). Fingerprinted single-molecule traces did not show specific mechanical features that can be unambiguously assigned to the ABD domain, although the final lengths of the single-molecule tethers are compatible with the ABD domain being unfolded at the end of the pulling protocol (Fig. 7g, h and Supplementary Fig. 6d). Indeed, in 15% of the force-ramp traces we did not detect any step other than those originating from I91 unfolding, suggesting that unfolding of ABD happens at forces below the ~10 pN detection limit of the AFM in those traces (Supplementary Fig. 6e). For the remaining traces, we detected a heterogeneous population of steps at 45 ± 30 pN (±SD) (Supplementary Fig. 6f), although some of these events probably are not related to ABD and originate from spurious, non-specific interactions with the surface. Taking all the single-molecule AFM results together, we conclude that the ABD domain has low mechanical stability, a biophysical property hallmark of other well-established protein mechanosensors, such as talin rod domains[59,63], β-catenin[64], or the kinase domain of titin[65]. This conclusion is further supported by coarse grained molecular dynamics simulations of the unfolding of ABD with a pulling force. We find low ABD unfolding forces not only when the pulling geometry is end-to-end (the same explored in the AFM experiments) but also when the domain is pulled through a subset of residues involved in binding to actin (Supplementary Fig. 6h–j). Taken together, these pieces of evidence support that c-Abl exhibits properties compatible with an intrinsic mechanosensing activity, which potentially couple directly mechanical stimuli to the modulation of its kinase output.

**FBP17 couples tension-induced PM and stress fiber remodeling.** Our results suggest that FBP17 is an important player in PM remodeling during the cell response to tension increase downstream of c-Abl. During this adaptation, in addition to changes in membrane curvature, stress fibers are reinforced[7]. The cross-talk between PM and the actin cytoskeleton is evident at many levels. Notably, caveolae are tightly linked to stress fibers downstream of c-Abl and mDia1[5]. Therefore, we hypothesized that FBP17 could be essential during the cross-talk between mechanosensitive membrane invaginations and stress fibers in the cell response to mechanical strain. Indeed, FBP17 deletion in Cav1 KO cells further increases their sensitivity to osmotic shock, suggesting that FBP17 also contributes to cell adaptation to tension changes

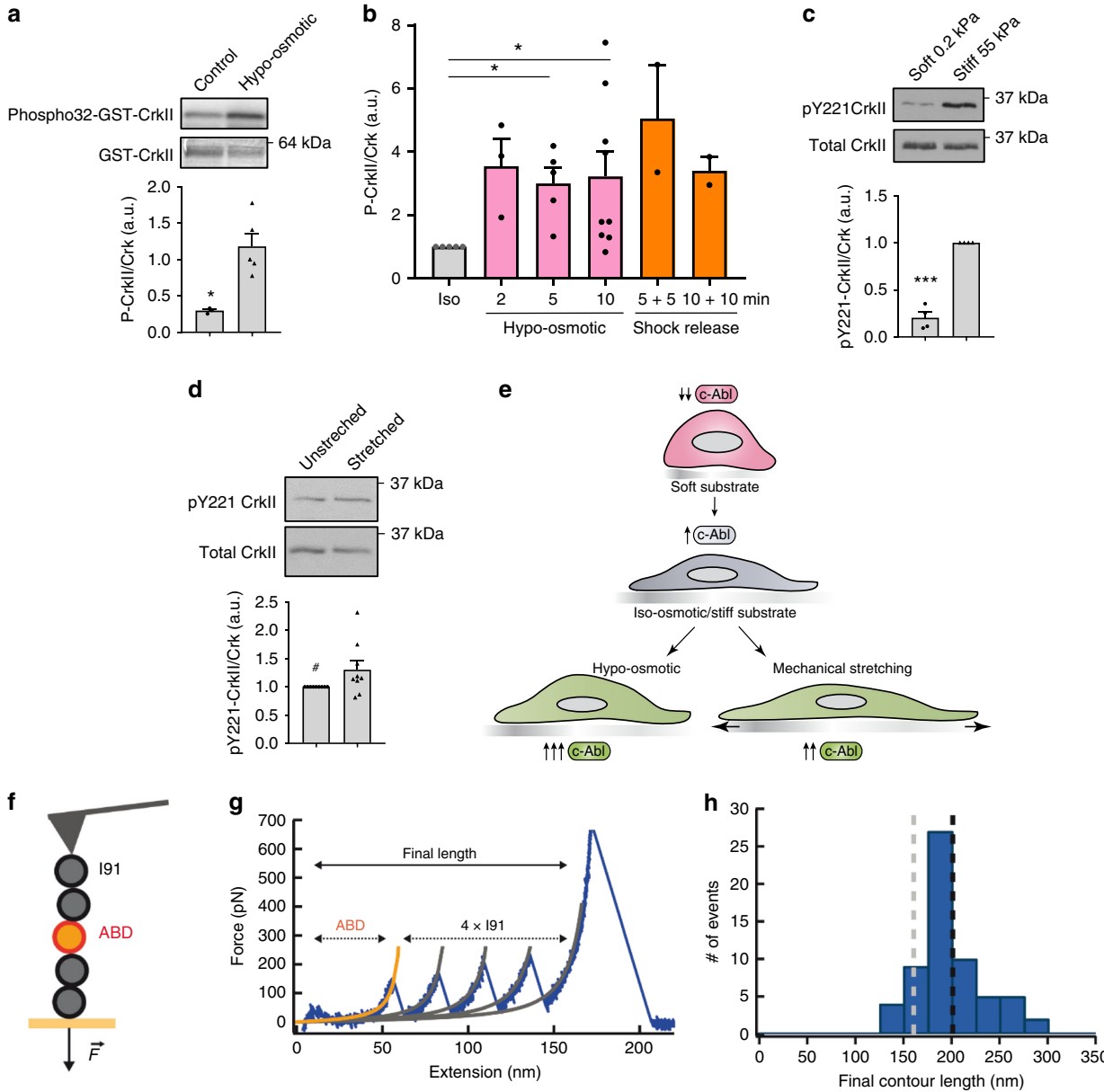

**Fig. 7 C-Abl kinase is activated by mechanical forces and its actin-binding domain is mechanosensitive. a** The kinase activity of endogenous c-Abl is elevated after hypo-osmotic treatment. Endogenous c-Abl was immunoprecipitated from cells treated with iso-osmotic or hypo-osmotic medium (60 mOsm, 10 min) and in vitro kinase assay was performed using GST-CrkII as substrate. Phosphorylation (radiolabeled) and the total protein (Coomassie stained) signals are shown. $N = 3$ (control) and $n = 5$ (hypo-osmotic) biologically independent samples from 3 independent experiments. Statistical analysis with a two-tailed unpaired $t$ test. *$P < 0.05$. **b** Kinetics of c-Abl activation upon osmotic swelling (30 mOsm) and after shock release is shown. Endogenous CrkII was immunoprecipitated from cells and pY221CrkII and CrkII was detected, normalized to iso-osmotic. From left to right, $n = 5, 3, 5, 9, 2,$ 2, from biologically independent samples from the corresponding independent experiments. Statistical analysis with a two-tailed unpaired $t$ test. *$P < 0.05$. **c** c-Abl kinase mechanosenses substrate rigidity. Endogenous CrkII was immunoprecipitated from cells grown in soft or stiff hydrogels and pY221CrkII and CrkII was detected. $N = 4$ biologically independent samples from 4 independent experiments. Statistical analysis with a two-tailed unpaired $t$ test. *$P <$ 0.05. **d** c-Abl kinase is activated by mechanical strain. Endogenous CrkII was immunoprecipitated from human fibroblasts unstretched and stretched for 10 min and pY221CrkII and CrkII was detected. #$P = 0.08$, $n = 9$ from biologically independent samples from 5 independent experiments. **e** Cartoon representing the different tensional states of a cell in culture and the activity of c-Abl depending on those tensional states. **f** Schematic representation of the heteropolyprotein (I91 ΔCys)$_2$-ABD-(I91 ΔCys)$_2$ used to characterize the mechanical properties of ABD. **g** Typical force-extension trace showing four unfolding peaks of the fingerprinting domain I91, which are preceded by a featureless extension which suggests that ABD unfolds at low forces. Solid lines show worm-like chain plots used to estimate contour lengths. **h** Final contour lengths of fingerprinted single-molecule force-extension traces. Dashed lines indicate theoretical final contour length values if ABD unfolds during the experiment (black) or remains folded (gray) are indicated. Data represent mean ± S.E.M.

and mechanoprotection through caveolae-independent mechanisms (Supplementary Fig. 4b).

FBP17 expression in cells reduces stress fibers[66] and binds mDia1 in vitro[38]. Furthermore, cip4, the only FBP17 family member in flies, inhibits Diaphanous actin polymerization activity[37]. Since density of stress fibers positively correlate with PM tension increase[7], stress fiber inhibitors ought to be blocked by tension. Several types of stress fibers have been characterized[7,67]. We quantified the effect of FBP17 on the different types of stress fiber[67,68] (Fig. 8a). FBP17 expression reduced dorsal stress fibers and transverse arcs and increased ventral stress fibers (Fig. 8b, c). This effect is consistent with the inactivation of mDia1, as silencing of mDia1 produces a similar phenotype[67,68]. In contrast, FBP17 3YE mutant did not modify dorsal fibers or transverse arcs, similar to GFP alone (Fig. 8b, c). Next, we studied the effect on purified proteins in an in vitro pyrene-actin polymerization assay (Supplementary Fig. 6k). Purified mDia1C (the active fragment of mDia1; ref. [69]) induced strong actin polymerization, as shown before (Supplementary Fig. 6l)[69]. Addition of FBP17 to mDia1C inhibited F-actin polymerization (Fig. 8d). However, FBP17 3YE mutant was unable to inhibit mDia1C, similar to control GST (Fig. 8d). Thus in vitro and in vivo analyses suggest that FBP17 inhibits mDia1-mediated actin polymerization and this is dependent on the tension-phosphorylated residues on the F-BAR domain.

To understand the role of FBP17 phosphorylation in the context of mechanoprotection, we studied the sensitivity of cells to hypo-osmotic shock in FBP17-silenced cells rescued with FBP17 or the phosphomimetic form (3YE). While FBP17 re-expression rescued the protection against osmotic swelling, the FBP17 3YE mutant, deficient for membrane bending and null for mDia1 inhibition, did not (Fig. 8e). Consistent with this, inhibition of c-Abl, which reduces FBP17 phosphorylation on those residues (Figs. 5 and 6d, e), had a protective effect (Fig. 8f). Together these results suggest that both activities of FBP17, membrane bending and stress fiber inhibition, are important to protect the cell against osmotic swelling.

## Discussion

The morphological changes in the PM and the cytoskeleton in response to mechanical strain are evident by imaging techniques[1,7,20], but the molecular mechanisms that underlie those changes and their functional consequences are not well understood. Here we identify a kinase-based tension-sensing pathway that controls the membrane-bending activity of FBP17 by directly phosphorylating the F-BAR domain in response to tension increase, which impedes its membrane-bending activity and stress fiber inhibition. This pathway ensures a coordinated response at the PM and cytoskeleton, allowing cell survival upon excessive tension. Upon tension increase, many molecular changes occur in the cell, which globally contributes to adapt the cell to the new mechanical properties of the surrounding matrix and/or cells. c-Abl kinase is activated by many stress responses[54] and here we show that it is also sensitive to mechanical stimuli. The biophysical properties of the c-Abl ABD domain could confer intrinsic mechanosensitivity to tension increase. The structural similarity between the ABD domains of c-Abl/Arg and well-characterized mechanosensitive domains, such as the R1 domain in Talin (R1 domain)[57,58,63] or those in FAK[70] or α-catenin[71], suggests that the ABD is a potential mechanosensing domain that couples mechanical stimuli with the regulation of c-Abl activity and its target substrates, such as FBP17. The current model for talin mechanosensitive properties suggests that the treadmilling of actin due to polymerization could unfold mechanosensitive domains[72]. Equivalent mechanical unfolding of the ABD domain

in c-Abl could lead to increased accessibility of the kinase active site. Indeed, the F-actin-binding properties of the ABD are essential to control c-Abl activity in vivo and in vitro[56], but how it does so remains unclear.

The activation of c-Abl by tension increase leads to the phosphorylation of the F-BAR domain of FBP17, which inhibits its membrane-bending activity and inhibition on mDia1 in cells and in vitro (Figs. 6 and 8). In vitro and in silico models provide evidence that tension increase reduces protein–protein interactions using N-BAR proteins as a model and tension reduction facilitates the tubulation activity of endophilin A1[73,74], whose tubulation activity is also regulated by phosphorylation in vitro[75], similar to other BAR proteins[48,76]. These studies raise the possibility that tension-driven regulation of BAR proteins may be a common mechanism used by cells to regulate the curvature of the membranes. Interestingly, in vitro assays using purified protein suggest that FBP17 itself is mechanosensitive[50]. A combination of mechanisms, i.e., intrinsic mechanosensitive properties and tension-triggered regulatory phosphorylation, could ensure that the membrane-bending activity is halted when tension at the PM is increased. Our results suggest that, under mild osmotic stimuli, the intrinsic mechanosensitive mechanism is not sufficient to inhibit the membrane-bending activity of FBP17, and a phosphorylation-based mechanism takes place (Fig. 6l, m). This suggests that depending on the degree of tension different mechanisms of transduction take place. Phosphorylation could act as an irreversible signal that completely inhibits FBP17 activity, as reversal of tension to basal conditions does not reduce the phosphorylation (Supplementary Fig. 5b).

Multiple FBP17 pools exist[50] and one of those pools is recruited to single caveolae, similar to other BAR proteins, such as pacsin2. While pacsin2 and 3 are needed for the correct formation of caveolae[16,17,47], FBP17 is not (Fig. 3). Interestingly, FBP17 is specifically required for the formation of caveolar rosettes, in accordance with the fact that, while the majority of the rosette-localized caveolae contain FBP17, only a small fraction of single caveolae are labeled positive for this protein (Fig. 2). Tension decrease could recruit FBP17 to caveolae, as actin polymerization inhibition, resulting in reduced tension, augments FBP17 colocalization with Cav1 (Fig. 1). This recruitment to single caveolae could favor the local inhibition of mDia1 by FBP17 (Fig. 8), as observed in Drosophila[37]. Inhibition of mDia1 is known to induce the formation of rosettes[5], which could explain the decrease of rosettes in FBP17-silenced or -deleted cells (Fig. 3).

Ensuring PM integrity is essential for cell survival and complex and diverse mechanisms exist to protect the PM. Tension relief appears to be an essential step in preventing membrane rupture[4] and the increased tension upon osmotic shock observed in FBP17-silenced cells suggest that this may be responsible for the reduced protection in these cells (Fig. 4). Interestingly, Cav1 and FBP17 double KO cells are more sensitive to osmotic swelling than Cav1 KO cells, suggesting that caveolae-independent pathways are regulated by FBP17. In agreement, FBP17 is relatively highly expressed in tissues where caveolae are not particularly abundant, such as the brain, while it is poorly expressed (in relative terms) in the skeletal muscle or lung[36], where caveolae are abundant. This suggests[36] that (i) FBP17 may have additional functions independent of caveolae[31,35,50] and/or (ii) that other F-BAR family members may play an analogous role in tissues where FBP17 is less abundant.

The actin cytoskeleton has been shown to be important to mechanoprotect the cell, although the exact mechanism is still unclear[8]. The fact that the FBP17 3YE mutant, which cannot inhibit mDia1, cannot protect cells from osmotic swelling (Fig. 8) suggests that stress fiber regulation is important for

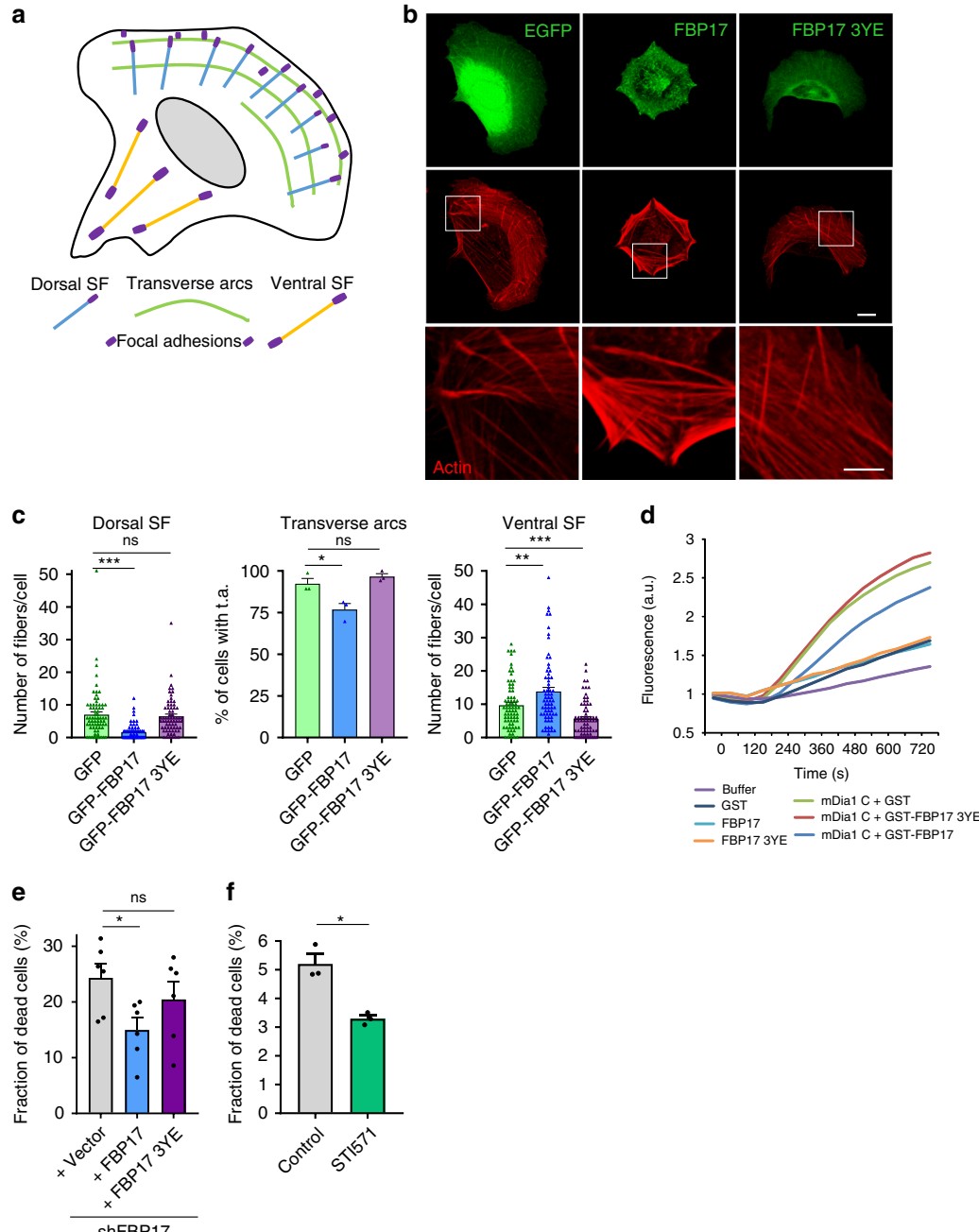

**Fig. 8 PM integrity and stress fiber inhibition are regulated by tension-regulated FBP17 phosphorylation. a** Cartoon depicting the different pools of stress fibers (SF). **b** Cells were transfected with plasmids expressing the indicated proteins and actin was stained. Scale bar 10 μm, boxes 5 μm. **c** Quantification of the different stress fiber pools in each condition as in **b**. $N = 74$ (GFP), $n = 68$ (GFP-FBP17), and $n = 68$ (GFP-FBP17 3YE) biologically independent cells from 3 independent experiments, for the quantification of dorsal and ventral SF; and $n = 3$ biologically independent samples from 3 independent experiments for the quantification of transverse arcs (t.a.). Statistical analysis with a two-tailed unpaired $t$ test. *$P < 0.05$; **$P < 0.01$; ***$P < 0.005$. **d** In vitro actin-pyrene polymerization regulated by mDia1C. Different proteins were incubated with mDia1C at 75 nM. Representative of three independent experiments. **e** FBP17 expression was silenced in human fibroblast and cells were treated with hypo-osmotic medium. Trypan blue-labeled cells were scored and the fraction of dead cells positive for trypan blue were calculated for each condition. $N = 6$ biologically independent samples from 6 independent experiments. Statistical analysis with a two-tailed unpaired $t$ test. *$P < 0.05$; ns non-significant. **f** Effect of inhibiting Abl kinases in cell sensitivity to osmotic shock. Cells were pretreated with vehicle or Abl inhibitor (STI571) for 30 min and then treated with osmotic shock (30 mOsm) for 10 min and dead cells were counted in each condition. $N = 3$ biologically independent samples from 3 independent experiments. Statistical analysis with a two-tailed unpaired $t$ test. *$P < 0.05$. Data represent mean ± S.E.M.

mechanoprotection. Similar to FBP17, EHD proteins are required for rosette formation, membrane stability, and stress fiber–caveolae linkage[15,27,77,78]. Additional studies will be needed to determine whether stress fibers, independent on caveolae, are important to mechanoprotect the cell.

Coordination of actin cytoskeleton and PM remodeling may be essential to mechanoprotect the cell from excessive mechanical insults. BAR proteins are the archetype of membrane bending and actin remodelers[28], and some of their family members, such as FBP17, may be specifically involved in controlling membrane

tension and mechanoprotection. Although *Saccharomyces cerevisiae* do not contain caveolae, they present PM invaginations, named eisosomes, that play an analogous mechanoprotective role[10]. Interestingly, various BAR domain proteins are responsible for the formation of these invaginations[79], raising the possibility that BAR domain-containing proteins were favored during evolution as the primary molecules needed to sense tension and ensure PM integrity.

## Methods

**Cell culture, infections, and transfections**. Cells were grown in Dulbecco's modified Eagle's medium (DMEM) and DMEM-F12 (for RPE-1) with 10% fetal bovine serum. Human foreskin fibroblasts were obtained from a healthy donor. U2os, RPE-1, HeLa, and 293T/17 were purchased from ATCC and were not authenticated. All cell lines tested negative for mycoplasma. Cell lines were kept at 37 °C in 5% $CO_2$ atmosphere. For lentiviral infections, 293T/17 cells were infected with packaging vectors and plasmids encoding the gene of interest. Supernatants were filtered (45 μm) and added to target cells. Plasmids were transfected using Fugene6 (Roche). In rescue experiments (Fig. 8e), human fibroblasts were infected with viruses encoding short hairpin RNAs (shRNAs) and hygromycin B-resistant cells were selected and infected with lentiviruses expressing FBP17 versions resistant to the shRNA sequence. siRNAs were transfected at 30 nM with oligofectamine (Invitrogen).

**Antibodies**. Rabbit-made antibodies: caveolin (BD, cat # 610060), Cav1 XP (Cell Signaling Technology, cat # 3267, immunofluorescence (IF) 1:350, western blotting (WB) dilution 1:10.000, I-EM 1:100), FBP17 (a kind gift from Pietro De Camilli[31]; for I-EM 1:25), FBP17 790 (Bethyl Laboratories, cat # A302–790A), Phospho-CrkII Y221 (Cell Signaling Technology, cat # 3491), pacsin2 (kindly provided by Richard Lundmark), SNX9 (Sigma-Aldrich, cat # HPA031410), and c-Abl (K12 Santa Cruz, cat # sc-131). Mouse monoclonal antibodies: CrkII (BD, cat # 610035), cip4 (BD, cat # 612556), c-Abl (8E9, BD, cat # 554148), anti-phosphotyrosine 4G10 (Millipore, cat # 05–321), α-tubulin (Sigma-Aldrich, cat # T-9026), GFP (Roche, cat # 11814460), Myc (9E10, Santa Cruz Biotechnology, cat # sc-40), and toca1 (a gift from Giorgio Scita). All antibodies were used at 1:100 for IF, 1:1000 for WB, and 1 μg/1 mg of lysate for immunoprecipitation, unless otherwise indicated. Uncropped and unprocessed WBs are shown in Supplementary Fig. 7.

**Reagents and plasmids**. siRNA oligonucleotides were purchased from Dharmacon and Ambion. As control non-targeting siRNAs, control #1 (Dharmacon) was used. siRNA oligonucleotides for FBP17 #1 sense sequence CCGAAUCAAUUGAUCAGAA, FBP17 #2 GGAUUUUGACGACGAGUUU, CIP4 GAACCUCAGUGUCCGUGUA, TOCA1 GGACGAACGAAGGACUAUU, SNX9 GGACAGAACGGGCCUUGAA, Pacsin2 CAAAUUAUGUGGAGGCGAU, and dynamin2[5]. LLOX shFBP17 hygro, a shRNA plasmid encoding GATCAGTTTGACAACTTA sequence against human FBP17, was previously reported[80] and kindly provided by Giorgio Scita. GFP-FBP17-expressing plasmid (pCAGGS FBP17) was kindly provided by Naoki Mochizuki[36]. Plasmids encoding GST-FBP17 (pGEX6P-1 FBP17) or Myc-FBP17 were kindly provided by Pietro De Camilli[31]. FBP17 from pGEX6P-1 FBP17 was subcloned into pEGFPC1 using BglII/SalI (sites were not maintained). The variants 3YE and 3YF (carrying Y190E, Y205E, Y287E or Y190F, Y205F, Y287F mutations, respectively, where positions are based on the variant Genebank BC143515) were generated by site-directed mutagenesis and cloned into pEGFPC1. pGEX6P-1 FBP17 3YF was obtained by site-directed mutagenesis. Bicistronic GFP-expressing lentiviral constructs expressing FBP17 wild-type and 3YE mutant shRNA-resistant cDNA variants (t27c, g30a, t33c, c36t, c39t, a42g) were obtained by directed mutagenesis and cloned into the BamHI/XhoI sites of the pRRL-IRES-EGFP. pEGFPC2 c-Abl was previously described[5] and GFP-WASH-expressing plasmid was kindly provided by Alexis Gautreau. pGEX-KText mDia1–549 (mDia1 C herein) was kindly provided by Henry N. Higgs[69]. The cDNA coding for the ABD of mouse c-Abl (amino acids 1019–1142, NM_001112703.2) was cloned between two I91 ΔCys domains of titin by iterative digestion/ligation steps using BamHI, BglII, and KpnI sites[61]. The final construct was cloned into pQE80L protein expression plasmid (Qiagen). The sequence of the final heteropolyprotein is shown in Supplementary Fig. 6g and can be obtained from Supplementary Data 1. Quantitative reverse transcription polymerase chain reaction primers are: FBP17 forward: TCAACATCCGCTTTTGTGACA, reverse: AAACTTTCACGATGGCCGTAA.

**ABD-based polyprotein purification**. Protein expression was induced in *Escherichia coli* BL21 (Gold) at OD600 = 0.6–1.0, using 0.4 mM IPTG, overnight at 20 °C. Soluble (I91ΔCys)$_2$-ABD-(I91ΔCys)$_2$ protein was purified by Ni-NTA and size exclusion (FPLC Superose 6 Increase 10/300 column from GE Healthcare Life Sciences) chromatographies[60]. Purification buffers included 10 mM dithiothreitol and the final buffer was 10 mM Hepes, pH 7.2, 150 mM NaCl, and 1 mM EDTA. Protein was stored at 4 °C until use.

**Hypo-osmotic treatment**. To test the sensitivity to membrane rupture (trypan blue-positive cells), attached cells were treated in the plate with the hypo-osmotic media to obtain 30 mOsm (1/10 dilution of the media) and incubated for 10 min. In some experiments (indicated in the figure legends), 60 mOsm and/or 2 min treatments were also conducted, in addition to the 30 mOsm and 10 min, for comparison purposes. Cells were collected and mixed with trypan blue at 0.02% w/v and positive/negative-stained cells were counted in a Neubauer chamber.

**Cell stretching**. Cells were grown for 24 h on silicone surfaces coated with pronectin and then subjected to 15–20% cyclic (0.5 Hz) uni-axial stretching for 1 h and 20 min (Fig. 4k), 30 min (Supplementary Fig. 3f), or 20 h (Fig. 4b) in a stretching device (Flexcell FX5000T System®). 0% of stretching was used as control.

**In vitro phosphorylation assay**. GST fusion proteins were purified using standard procedures. After elution, GSTs were dialyzed against 10 mM $MgCl_2$ and 20 mM Tris-HCl pH 7.4 and 0.2–0.8 μg were incubated with recombinant c-Abl kinase fragment (New England Biolabs) or full-length c-Abl (Life Technologies) in reaction buffers as indicated by the manufacturers. When wild type and 3YF FBP17 were compared, the proteins were kept in the agarose beads and incubated with the kinase after washing in kinase buffer. Reactions were incubated in the presence of [γ-$^{32}$P] ATP at 30 °C for 60 min. For mass spectrometry, the reaction was scaled up sixfold and the GST fusion substrate was kept in glutathione-agarose beads. Phosphorylation reactions were stopped by extensive washing with cold 100 mM ammonium bicarbonate and processed for mass spectrometry.

**Reporting summary**. Further information on research design is available in the Nature Research Reporting Summary linked to this article.

## Data availability
Data supporting the findings of this manuscript are available from the corresponding author upon reasonable request.

## Code availability
Custom software (written in Igor Pro 6) is available upon reasonable request.

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

## Acknowledgements

We thank Pietro De Camilli, Giorgio Scita, Richard Lundmark, Naoki Mochizuki, Alexis Gautreau, and Henry Higgs for valuable reagents; Robert B. Best for sharing computing materials; Miguel Sánchez for text editing; and Milagros Guerra Rodríguez and Germán Andrés Hernández (Centro de Biología Molecular Severo Ochoa) for technical assistance in EM experiments. This study was supported by grants from the Spanish Ministry of Economy, Industry and Competitiveness (MINECO)/Agencia Estatal de Investigación (AEI)/European Regional Development Fund (ARDF/FEDER) "A way to make Europe" Grants (MINECO; SAF2011–25047, SAF2014–51876-R, SAF2017–83130-R, IGP-SO grant MINSEV1512–07–2016, CSD2009–0016 and BFU2016–81912-REDC), Fundació La Marató de TV3 (674/C/2013), and the Worldwide Cancer Research Foundation (♯15–0404), all to M.A.D.P. M.G.-G. is sponsored by a FPU fellowship (FPU15/03776). This project has received funding from the European Union's Horizon 2020 research and innovation programme under the Marie Sklodowska-Curie grant agreement No. 641639. D.D.S. is supported by grants PGC2018–099321-B-I00 and RYC-2016–19590 from the Spanish Ministry of Science, Innovation and Universities. J.A.-C. acknowledges funding from MINECO grants, BIO2017–83640-P (AEI/FEDER, UE) and RYC-2014–16604. J.A.-C. and M.A.D.P. are members of the Tec4Bio consortium (ref. P2018/NMT4443; "Actividades de I+D entre Grupos de Investigación en Tecnologías," Comunidad Autónoma de Madrid/FEDER, Spain). C.H.-L. is recipient of an FPI predoctoral fellowship (BES-2015–073191). C.L. is supported by institutional grants from the Curie Institute, INSERM, and CNRS and by grants from Association Française contre les Myopathies (CAV-STRESS-MUS no. 14293), Agence Nationale de la Recherche (MOTICAV ANR-17-CE13–0020–01), the Fondation ARC pour la Recherche sur le Cancer (Programme Labellisé PGA1-RF20170205456), and programme ECOS no. C17S03. R.G.P. was supported by the National Health and Medical Research Council (NHMRC) of Australia (program grant, APP1037320 and Senior Principal Research Fellowship, 569452) and the Australian Research Council Centre of Excellence (CE140100036). We acknowledge the Australian Microscopy & Microanalysis Research Facility at the Center for Microscopy and Microanalysis at The University of Queensland. The CNIC is supported by the Instituto de Salud Carlos III (ISCIII), the Ministerio de Ciencia, Innovación y Universidades (MCNU), and the Pro CNIC Foundation and is a Severo Ochoa Center of Excellence (SEV-2015–0505).

## Author contributions

A.E. and M.A.D.P. conceived/supervised the project, designed experiments, analyzed results, and wrote the manuscript. M.G.-G., C.V.-L., D.D.S., J.A.-C., C.L., and R.G.P. designed experiments, edited the text, and/or contributed to the writing. A.E., D.M.P., S.S., M.G.-G., E.C., C.H.-L., D.V.-C., C.V.d.L., N.A., A.L.-C., R.S., and D.D.S. designed, performed, and analyzed experiments.

## Competing interests

The authors declare no competing interests.
