## [Peer Review File · Nature Communications]

REVIEWERS' COMMENTS:

Reviewer #1 (Remarks to the Author):

This is a significantly improved version of a manuscript reporting how mechanosensitive interplay between c-Abl and FBP17 contributes to mechanoadaptation. The authors have satisfactorily addressed my previous concerns, and I have only few minor suggestions for revising the manuscript.

1. Legend to Fig. 1e is slightly confusing, and could be clarified. What do – and + present in the figure? What are the numbers of cells and spots analyzed here?

2. There are several typos in the text, e.g. 'Discussion' line 705 (... which impedes its-membrane bending...); line 707 (Upon increase tension many...); legend to Fig. 6d-e (...were expressed in hela cells and....); legend to Fig. 7b (Kinetic of c-ABL activation....).

Reviewer #4 (Remarks to the Author):

In this manuscript the authors present comprehensive data sets demonstrating a novel Abl-FBP17 mechanical sensing model that couples plasma membrane curvature and stress fiber remodeling during mechanical stress adaptation. Various state-of-the-art experimental techniques and working systems are applied. The conclusions are well supported. These results will help to understand how mechanical forces are translated into biochemical signals that coordinate the structural changes observed at the plasma membrane and the underlying cytoskeleton during mechanical stress. It could be a highly cited report in the future.

It looks like FBP17-mediated mechanical sensing response could be a general mechanism in many cell types. In this manuscript human fibroblast and other cultured cell lines were utilized, most of them are cancer or cancer-like cell lines. Caveolae is widely expressed in many tissues and cell types, however, it is highly abundant in some specific organs, such as lung, fat and muscle. In contrast, some of other tissues have limited expression level. Although experimentally this may be beyond the scope of this manuscript, it will be very helpful to address the physiological relevance of this novel pathway in tissue/cell type specific manner, at least in the section of discussion. Does FBP17 have the similar tissue distribution pattern as caveolae?

Minor points: In some western blot figures, there are lines between the bands. Does this mean the results are from different blots? If possible, the bands need to be run in the same gel, so the proper comparisons can be made.

Point-by-point rebuttal letter

Reviewers' comments:

Reviewer #1 (Remarks to the Author):

"This is a significantly improved version of a manuscript reporting how mechanosensitive interplay between c-Abl and FBP17 contributes to mechanoadaptation. The authors have satisfactorily addressed my previous concerns, and I have only few minor suggestions for revising the manuscript. "

"1. Legend to Fig. 1e is slightly confusing, and could be clarified. What do - and + present in the figure? What are the numbers of cells and spots analyzed here?"

We have rewritten the figure legend to make it more understandable and we specify the meaning of "-" and "+". We also include the number of cells and spots analyzed, as requested. *"2. There are several typos in the text, e.g. "Discussion" line 705 (... which impedes its-membrane bending...); line 707 (Upon increase tension many...); legend to Fig. 6d-e (...:were expressed in hela cells and...); legend to Fig. 7b (Kinetic of c-ABL activation...)."*

We are sorry for these mistakes. We have corrected these typos. Thanks for pointing them out. These are the changes we identified and corrected:

..which impedes its membrane bending → "...which impedes its membrane bending..."

-"Upon increase tension many molecular changes occur in the cell that globally contributes to adapt" → "Upon increase tension many molecular changes occur in the cell that globally contribute to adapt..."

- were expressed in hela → "...were expressed in HeLa..."

- "Kinetic of c-ABL activation" → "...Kinetics of c-Abl activation..."

Reviewer #4 (Remarks to the Author):

In this manuscript the authors present comprehensive data sets demonstrating a novel Abl - FBP17 mechanical sensing model that couples plasma membrane curvature and stress fiber remodeling during mechanical stress adaptation. Various state-of-the-art experimental techniques and working systems are applied. The conclusions are well supported. These results will help to understand how mechanical forces are translated into biochemical signals that coordinate the structural changes observed at the plasma membrane and the underlying cytoskeleton during mechanical stress. It could be a highly cited report in the future.

We thank the reviewer for her/his comments.

It looks like FBP17-mediated mechanical sensing response could be a general mechanism in many cell types. In this manuscript human fibroblast and other cultured cell lines were utilized, most of them are cancer or cancer-like cell lines. Caveolae is widely expressed in many tissues and cell types, however, it is highly abundant in some specific organs, such as lung, fat and muscle. In contrast, some of other tissues have limited expression level. Although experimentally this may be beyond the scope of this manuscript, it will be very helpful to address

the physiological relevance of this novel pathway in tissue/cell type specific manner, at least in the section of discussion. Does FBP17 have the similar tissue distribution pattern as caveolae?

We now describe this point in the discussion (page 15, line 453...). The expression pattern of FBP17 and the tissue abundance of caveolae do not always correlate. We discuss that this suggest: i) FBP17 may have caveolae independent functions, as our and other studies suggest (cited in the Discussion), and/or ii) other F-BAR family members may compensate the low expression of FBP17 in cells with abundant caveolae.

Minor points: In some western blot figures, there are lines between the bands. Does this mean the results are from different blots? If possible, the bands need to be run in the same gel, so the proper comparisons can be made.

The lines indicate that between the two lanes divided by the line there are additional lanes. These lanes were cropped, as they were unrelated or not informative for the particular experiment. Importantly, the shown bands correspond to the same gel and to the same exposition. We now include the original full western blots where the lanes of interest are marked (Supplementary Figure 7).